# Determination of Sulphonamides and Tetracycline Residues in Liver Tissues of Broiler Chicken Sold in Kinondoni and Ilala Municipalities, Dar es Salaam, Tanzania

**DOI:** 10.3390/antibiotics11091222

**Published:** 2022-09-08

**Authors:** Winstone J. Ulomi, Fauster X. Mgaya, Zuhura Kimera, Mecky I. Matee

**Affiliations:** 1Tanzania Bureau of Standards, P.O. Box 9524, Dar es Salaam 16103, Tanzania; 2Department of Microbiology and Immunology, Muhimbili University of Health and Allied Sciences, P.O. Box 65001, Dar es Salaam 11103, Tanzania

**Keywords:** antimicrobial residues, tetracycline, sulphonamides, maximum residual limit, acceptable daily intake, broiler chicken, liver tissues

## Abstract

In Tanzania, the increased demand for animal-derived foods, particularly eggs, meat, and milk, has resulted in the intensification of farming systems with the use of antimicrobials, particularly sulphonamides and tetracyclines. According to the FAO/WHO Codex Alimentarius commission, concentrations of antimicrobial residues in food exceeding the acceptable daily intake (ADI) and maximum residual limit (MRL) pose a health risk to consumers. This cross-sectional study determined the concentrations of sulphonamide and tetracycline residues in the liver tissues of commercial broiler chicken sold in Dar es Salaam, Tanzania, to find out whether the amounts of residues were within the legally permitted and acceptable limits in food. We conveniently sampled eighty-four liver tissue samples from broiler chicken sold in two out of six large markets in Dar es Salaam. The amounts of tetracycline and sulphonamide residues were determined using an ELISA kit (Shenzhen Lvshiyuan Biotechnology Company, Shenzhen, China). The results showed that all 100% (*n* = 84) samples contained tetracycline residues and 21.4% (*n* = 18) samples contained sulphonamide residues, while 21.4% (*n* = 18) contained both sulphonamide and tetracycline residues. The concentrations of sulphonamide residues were within the maximum residual limit (MRL). However, 90.5% (*n* = 76) of the samples had tetracycline levels that exceeded the acceptable daily intake (ADI) range 0–3 µg/kg and 13.1% (*n* = 11) of the samples had tetracycline levels that exceeded the maximum residue limit of 300 µg/kg. The observed presence of antibiotic residues in the poultry tissues poses a health risk to consumers, and may lead to antimicrobial resistance micro-organisms, which may spread to humans and animals via the environment. Vigorous surveillance and observation of the withdrawal periods should be advocated to ensure that the food from animals is safe with regard to the residues of veterinary medicines.

## 1. Introduction

Antimicrobial use in food animals, particularly short-cycled species such as pigs and poultry, is a global concern [1]. This is due to the use of intensified farming, which has been linked with antimicrobial use (AMU) and the emergence and spread of antimicrobial resistance (AMR) pathogens [2]. In Tanzania, pork consumption is projected to rise from 42,700 to 170,000 metric tons between 2017 and 2030 [2,3]. Similar estimates show that the annual chicken meat production is projected to rise from 22,000 tons in 2017 to 37,200 tons in 2022 [3,4]. The most commonly used antimicrobials in livestock production in Tanzania include tetracycline, sulphonamides, penicillin, aminoglycosides and macrolides [3,4,5]. A recent review by Kimera et al. [6] revealed that the percentage of farms using antibiotics in poultry production in Tanzania was 100%. Tetracycline is the most used antibiotic, followed by sulphonamides, penicillin, and aminoglycosides [4,7]. This is in line with a review conducted by Azabo et al. [7] that reported tetracycline to be the most consumed (8,057,240 kg) antibiotic in animal farming, followed by sulphonamides and trimethoprim (3,057,240 kg) [7]. Tetracyclines and sulphonamides are the most commonly used antibiotics in poultry production in Tanzania and other African countries [3,4,6]. These antibiotics are readily available from veterinary shops and vendors and are accessed easily, often without restrictions [4,7]. Informal vendors sell these antibiotics at informal markets and along the road, without any prescription being required or restrictions imposed [5,8]. Previous studies have shown that most farmers have inadequate knowledge and inappropriate attitudes and practices regarding the judicious use of antibiotics, posing serious human and animal health issues [5,9].

The two international metrics that are used to monitor antimicrobial residues in animal food include the maximum residue limit (MRL), which refers to the maximum allowable level of a residue, and the acceptable daily intake (ADI), which refers to the acceptable amount of a veterinary drug expressed on a body weight basis that can be ingested daily over an entire human lifetime without any appreciable health risk [10]. The MRLs and ADIs are set by the Codex Committee on Pesticide Residues (CCPR), based on recommendations made by the FAO/WHO Joint Meeting on Pesticide Residues (JMPR) [10]. 

According to the regulations set by the European Union and the Codex Alimentarius Commission, the MRLs in liver tissue are 100 µg/kg for sulphonamides and 600 µg/kg for tetracycline, while the acceptable daily intake (ADI) ranges for tetracycline and sulphonamides are 0–30 µg/kg and 0–50 µg/kg, respectively [10]. 

The methods for the detection of antibiotics residues can be grouped as screening and confirmatory [11]. 

Several screening and confirmatory analytical methods for the detection of antibiotic residues have been developed, including thin-layer chromatography [12], the enzyme-linked immunosorbent assay (ELISA) [13], the Nouws antibiotic test (NAT), commercial ampoule tests, and the Premi test [14]. Recent studies have shown that that the enzyme-linked immunosorbent assay (ELISA) method has sensitivity of 90% and specificity of 80% in determining antibiotic residues in meat samples [15,16]. 

The enzyme-linked immunosorbent assay (ELISA) method is less time-consuming and cheaper than high-performance liquid chromatography (HPLC) and mass spectrophotometry (MS) [6,9,17,18,19], and does not offer a cost-effective screening platform, particularly in resource-limited settings. Indeed, recent studies have shown that using an ELISA can yield results that are comparable to those from the use of HPLC/MS. The remaining antibiotics were found in broiler chicken tissues using the ELISA method in a recent study by El Tahir et al. 2021.

We conducted this study in Dar es Salaam, a commercial city in Tanzania, where the proportion of farmers using antimicrobials in poultry farming is 100%, which is the most intense rate in the country [20,21]. The aim was to determine the concentrations of tetracycline and sulphonamide residues in the liver tissues of broiler chicken in comparison with the maximum residue limit (MRL) and acceptable daily intake (ADI). The choice of liver tissues was based on the results of a study that showed that the residues of different antibiotics were higher in the liver than in the meat and that antibiotics remain for longer periods in liver tissues, especially if the initial dosage of the antibiotic is high or if the withdrawal time is not sufficient before slaughtering [16]. Furthermore, broiler chicken livers and other internal parts are heavily consumed by the local population in Dar es Salaam, and would, thus, pose a public health concern.

## 2. Results

### 2.1. Percentage of Broiler Chicken Liver Tissue with Tetracycline and Sulphonamide Residues

As shown in Table 1, all liver samples were positive for tetracycline residues and 21.4% (*n* = 18) were positive for sulphonamides. For the sulphonamides, 65.5% (*n* = 55) were negative and 13.1% (*n* = 11) were undetectable. 

**Table 1 antibiotics-11-01222-t001:** The proportions of liver tissues positive for sulphonamides and tetracycline residues in Shekilango and Manzese markets.

Sample Site	Drugs	* Positive (%)	* Negative (%)	* Not Detected (%)
Shekilango	Tetracycline	42 (100)	0 (0)	0 (0)
	Sulphonamides	14 (33.33)	22 (52.38)	6 (14.29)
Manzese	Tetracycline	42 (100)	0 (0)	0 (0)
	Sulphonamides	4 (9.52)	33 (78.57)	5 (11.91)

Note: * positive = ≥4 µg/kg, negative = <4 µg/kg, not detected = <0.1 µg/kg for tetracycline; * positive = ≥1 µg/kg, negative = <1 µg/kg, and not detected = <0.1 µg/kg for sulphonamides. As shown in Figure 1, eighteen (21.4%) samples had dual drug residues, of which 4.76% (*n* = 4) were from the Manzese market, while 16.67% (*n* = 14) were from the Shekilango market.

### 2.2. Concentration of Sulphonamides and Tetracycline Residues in Broiler Chicken Liver Tissues

Table 2 shows the median concentration of tetracycline and sulphonamide residues and the dispersion (interquartile range) of each of the tested drug residues. The highest and lowest concentrations for tetracycline were found to be 500 µg/kg and 10.25 µg/kg, while for sulphonamides the concentrations were 1.65 µg/kg and 1.00 µg/kg, respectively.

### 2.3. Concentrations of Tetracycline and Sulphonamides versus the Recommended Maximum Residue Limit (MRL)

Of all 84 tested liver tissue samples, 11 (13.1%) samples had drug residue levels above the MRL for tetracycline, while none of the samples had drug residue levels above the maximum residue limit for sulphonamides. The highest concentration of tetracycline obtained was 500 µg/kg, which was approximately 67% above the standard MRL. The different concentrations and MRL cut-off points are shown in Figure 2.

### 2.4. Concentrations of Sulphonamides and Tetracycline versus Acceptable Daily Intake (ADI)

As shown in Figure 3, 90.5% (*n* = 76) of the 84 tested samples were found to have tetracycline concentrations above the acceptable daily intake, while all positive samples for sulphonamides were within the acceptable daily intake range. 

## 3. Discussion

The results from the study show that all tested samples contained tetracycline residues (100%; *n* = 84), 21.4% (*n* = 18) contained sulphonamides, while 18 (21.4%) samples contained both tetracycline and sulphonamide residues. This finding is in line with another study on poultry production in Dar es Salaam, which reported tetracycline to be the most used antibiotic at 49%, followed by sulphonamides at 18% [7], which does indicate non-observation of the withdrawal period [4]. According to the FAO/WHO under the Codex Alimentarius commission, the concentrations of both tetracycline and sulphonamide residues were within the maximum residual limit (MRL) range [10], which indicates that the amount of residues was at the legally permitted or acceptable levels in food. However, 76 (90.5%) liver samples had tetracyclines residues exceeding the acceptable daily intake (ADI) range of 0–3 µg/kg and 11 (13.1%) liver samples had tetracycline residues exceeding the maximum residue limit of 300 µg/kg, an index developed by the Joint FAO/WHO Expert Committee on Food Additives (JECFA). Unlike the MRL, the ADI is a reference to the amount that can be ingested over a lifetime, without any appreciable health risk. This implies that over time the concentration of tetracycline residues observed in the poultry meat will cause a health risk [22,23,24]. The presence of antibiotic residues in poultry has been associated with the emergence and presence of micro-organims that are resistant to drugs [7,16,25,26]. Indeed, in recent studies conducted in poultry slaughter slabs in Dar es Salaam, raw meat and cloaca swabs revealed that 69.3% of *Escherichia coli* samples were resistant to multiple drugs (MDR) and very highly resistant to tetracycline (91.9%) [25]. Due to the spread of resistomes, the antimicrobial resistance genes (ARGs) can be spread from animals to humans directly or via the environment, thereby acting as a driver of AMR spread in the community [22,25,27].

Our findings on sulphonamide residues differ from those from another study conducted in commercial layer chicken eggs in Dar es Salaam, which reported that 29.2% of layer chicken egg samples having sulphonamide residues above the maximum residue limit, possibly due to the technologies used [9]. Nonetheless, both studies seem to indicate the presence of antibiotic residues in poultry products, posing significant health risks to humans and animals, as well as resulting in environmental contamination with ARGs, which foster the further spread of resistance, causing significant economic losses [7]. Unfortunately, an overwhelming proportion of human and veterinary medicines sold in pharmacies are dispensed without a prescription [5,8], including prohibited antimicrobials such as furazolidone [21]. Moreover, for the great majority of poultry farmers, the treatment services are provided by veterinary paraprofessionals, the majority of whom (72.5%) have not received any formal training on AMU or AMR [28,29]. This situation calls for vigorous monitoring and observation of the withdrawal periods, which should be advocated to ensure that the food from animals is safe with regard to the residues of veterinary medicines.

Fortunately, Tanzania has policies and guidelines that address the use of antimicrobials in animal food and agricultural systems using a ‘one health’ approach [30]. However, these policies and regulations are poorly enforced due to weak systems, particularly in animal food production [22,28,29]. 

We, thus, call upon the government of Tanzania through its regulatory agencies to enforce the national regulatory framework for antimicrobial use (AMU) in animal food production. This should include limiting or restricting the use of antimicrobials in animal farming, focusing on promoting responsible antimicrobial use via the existing guidelines. The continuous monitoring of AMU in animal food production and community sensitization to the observation of withdrawal periods should be highly advocated. Furthermore, the use of antimicrobials as growth promoters should be banned and infection prevention and control measures such as biosecurity and vaccinations should be highly advocated. We further recommend a ‘one health’ approach, as embraced in Tanzania’s AMR National Action Plan 2017–2022 to promote rational use and reduce antimicrobial consumption and resistance in humans and animals [30]. Finally, veterinary paravets, who are actually providing veterinary services to most farmers [28,29], should be trained on the judicious of antimicrobials, as well as receiving refresher courses on the diagnosis and prevention of infections through short courses arranged by The Veterinary Council of Tanzania.

This study does provide a useful insight on the presence and concentration of sulphonamide and tetracycline residues in poultry meat in comparison with two international indices, namely the maximum residual limit (MRL) and acceptable daily intake (ADI). However, we recognize the limitations of this study. Sulphonamide and tetracycline are not the only antimicrobials used in poultry farming, meaning this study underestimates the actual human and animal health risks as well as the environmental hazards due to other antimicrobial residues present in poultry meat sold in Dar es Salaam markets. The JECFA recommended an MRL of 300 µg/kg for tetracycline and its classes in liver tissue. In this study, the four epimers of tetracycline were not included in the analysis, which is a limitation. This study also involved only two of the six major slaughter centers, namely Shekilango and Manzese.

## 4. Material and Methods

### 4.1. Study Area 

The study area was Dar es Salaam, which is the largest and the main commercial city in Tanzania, located in the coastal region at a latitude of 6.792° S and a longitude of 39.2083° E. The city has a number of slaughtering markets, which include Manzese, Shekilango, Kisutu, Mtambani, Stereo, and Magomeni.

### 4.2. Study Design

The study was a cross-sectional study conducted from November 2021 to April 2022. The Shekilango and Manzese slaughtering slabs in Ubungo municipality were purposefully selected for being among the largest in Dar es Salaam. The average numbers of chickens slaughtered in Shekilango and Manzese slaughtering markets equal 3500 and 2500, respectively (estimates from the market authority).

The actual sample collection and processing took only one month in November 2021. 

### 4.3. Sampling Technique

All broiler chicken brought to the two markets during the study period were eligible for inclusion. The chickens included in the study were randomly sampled from the farmers or brokers before taken for slaughtering. Before slaughter, close monitoring was observed to avoid mixing with other chickens.

### 4.4. Specimen Collection and Transportation

About 10 g of liver tissue was collected from each sampled chicken using clean forceps, which was then immediately placed into a labeled plastic bag and put into an ice cold box at 8 °C to 10 °C. A total of 84 liver samples were collected, with 42 liver tissues from each slaughter market. The samples were transported to the laboratory within 3 h of collection and stored at −20 °C prior to further processing. 

### 4.5. Chemicals and Reagents

The following chemicals reagents and procedures were used.

For tetracycline: 6× standard solution (1 mL each) (0 ppb, 0.1 ppb, 0.3 ppb, 0.9 ppb, 2.7 ppb, and 8.1 ppb), enzyme conjugate (7 mL), antibody working solution (7 mL), substrate A (7 mL), substrate B (7 mL), stop solution (7 mL), 20× sample extract A (15 mL), 2× sample extract B (100 mL), 20× sample diluent (10 mL), 1 M NaOH solution.

For sulphonamides: 6× standard solution (1 mL each) (0 ppb, 1 ppb, 3 ppb, 9 ppb, 27 ppb, 81 ppb), enzyme conjugate (7 mL), antibody working solution (7 mL), substrate A (7 mL), substrate B (7 mL), stop solution (7 mL), 20× concentrated washing buffer (40 mL), 20× concentrated re-dissolving solution (50 mL), ethyl acetate, N-hexane.

### 4.6. Extraction Procedure

The samples were homogenized according to the instructions from the manufacturer (Shenzhen Lvshiyuan Biotechnology Company, Shenzhen, China) for each antibiotic kit using a meat mincer [31,32]. 

For sulphonamides, a sample of 2 ± 0.05 g of tissue homogenate (as specified on the kit) was weighed in a 50 mL centrifuge tube and 6 mL of ethyl acetate was added to that. After shaking for 2 min, the resultant homogenate was centrifuged at 4000 rpm at 15 °C for 10 min. Approximately 3 mL clear organic phases were taken into a dry container and blown until dry with air completely using a water bath at 50–60 °C. The dry residues were dissolved in 1 mL of the dilute redissolving solution, then in 1 mL of N-hexane (Loba Chemie Pvt. Ltd., Jehangir Villa, 107, Wodehouse Rd., Colaba, Mumbai, India), then they were mixed for 30 s and centrifuged at 4000 rpm at 15 °C for 5 min. The upper layer of the N-hexane phase was removed and 50 μL of deposit was taken for analysis.

For tetracyclines, a sample of 2 ± 0.05 g of the homogenized tissue (as specified on the kit) was weighted in a 50 mL centrifuge tube, then 3 mL of diluted sample extract a was added and shaken for 3 min. Then, 600 μL of 1 M NaOH (Loba Chemie Pvt. Ltd., Jehangir Villa, 107, Wodehouse Rd., Colaba, Mumbai, India) and 2.4 mL of diluted extract were added and shaken for 3 min. The mixture was then centrifuged at 4000 rpm at room temperature (20–25 °C) for 5 min. About 50 μL of supernatant was added to the 450 μL of diluted sample diluent and mixed evenly, then 50 μL was taken for the tetracycline residue analysis.

### 4.7. Competitive Enzyme Linked Immunosorbent Assay

The antibiotic competitive enzyme immunosorbent assay kits were purchased from Shenzhen Lvshiyuan Biotechnology Company, China, for tetracyclines (TCs) (catalog No. LSY-10006) and sulphonamides (SAs) (catalog no. LSY-10009). Two extractions were performed for each sample. The sensitivity, detection limit, and recovery rate for each kit were as follows: tetracycline (TC) (sensitivity of 0.1 ppb, detection limit of 4 ppb, recovery rate of 100 ± 20%, linear range of 0 ppb to 8.1 ppb, and stability at 25 °C); sulphonamides (SAs) (sensitivity of 1 ppb, detection limit of 1 ppb, recovery rate of 85 ± 25%, liner range of 1 ppb to 20 ppb, and stability at 25 °C). Each immunoassay kit contained sufficient material for 96 measurements. Each microtiter consisted of 96 wells, pre-coated with coupling antigens for each antibiotic. 

The procedure from the manufacturer (Shenzhen Lvshiyuan Biotechnology Company, Shenzhen, China) for each antibiotic kit was followed. The procedure includes a list of different compounds from the sulphonamide class as well as compounds from the tetracycline class and their corresponding cross-reactivity. In brief, each well received 50 μL of each standard solution or prepared sample, followed by 50 μL of anti-antibiotic solution. The plates were incubated at room temperature for 1 h before being washed three times with 250 μL of washing buffer and 100 μL of enzyme conjugate added to each well. After 15 min, the plates were washed three times with 250 μL of washing buffer. Then, 50 μL of substrate and 50 μL of chromogen were added to each well and incubated at room temperature for 15 min in the dark. Finally, 100 μL of stop solution was added to stop the reaction, and the absorbance at 450 nm was measured within 30 min.

Table 3 below shows the different compounds belonging to sulphonamides and tetracycline that the ELISA kits were capable of testing and the cross-reactivity for each compound. 

### 4.8. Statistical Analysis

The data were analyzed using Green Spring Excel analysis software (data analysis of food safety ELISA test, duplicate samples; Shenzhen Lvshiyuan Biotechnology, Shenzhen, China) and Graph Pad Prism version 9.0.0 for Windows (Graph Pad Software, San Diego, CA, USA, www.graphpad.com, aeecssed on 20 November 2021). The concentrations of sulphonamides and tetracycline were summarized as medians and interquartile ranges, since they were not normally distributed. Individual plot graphs were used to compare the concentrations of drug residues in the samples with the recommended maximum residual limit. The violin graph was used to compare the concentrations of sulphonamides and tetracycline with the acceptable daily intake (ADI). The proportions of samples with positive concentrations of antibiotic were calculated and plotted in a compound bar chart. The proportion of samples with dual drugs was presented in a pie chart.

The limits for the detection and quantification were set in Green Spring Excel analysis software (data analysis of food safety ELISA test, duplicate samples), and the judgmental values (Shenzhen Lvshiyuan Biotechnology Company, Shenzhen, China) were as follows: ** positive = ≥4 µg/kg, negative = <4 µg/kg, and not detected = <0.1 µg/kg for tetracycline; * positive = ≥1 µg/kg, negative = <1 µg/kg, and not detected = <0.1 µg/kg for sulphonamides.*

### 4.9. Ethical Consideration

Ethical approval was obtained from Muhimbili University of Health and the Allied Science Institute Research Board (DA.282/298/01.C/1031). Permission to conduct the study was sought from municipal officials and Shekilango and Manzese market authorities. 

## 5. Conclusions

This study found tetracycline residues in all liver samples and sulphonamide in 21.4% (*n* = 18) of samples, while 21.4% (*n* = 18) had both sulphonamide and tetracycline residues. The concentrations of sulphonamide residues were within the maximum residual limit (MRL). However, 90.5% (*n* = 76) of samples had tetracycline levels that exceeded the acceptable daily intake (ADI) range of 0–3 µg/kg, and 13.1% (*n* = 11) samples had tetracycline levels that exceeded the maximum residue limit of 300 µg/kg, the amount that can be ingested over a lifetime without any appreciable health risk, calling for regulatory bodies to take immediate action.

## Figures and Tables

**Figure 1 antibiotics-11-01222-f001:**
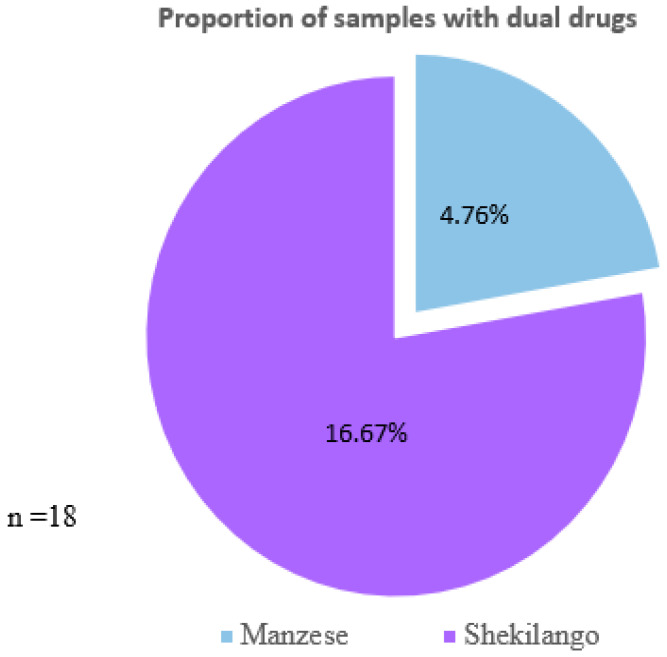
Proportion of samples with dual drug residues from Manzese and Shekilango markets.

**Figure 2 antibiotics-11-01222-f002:**
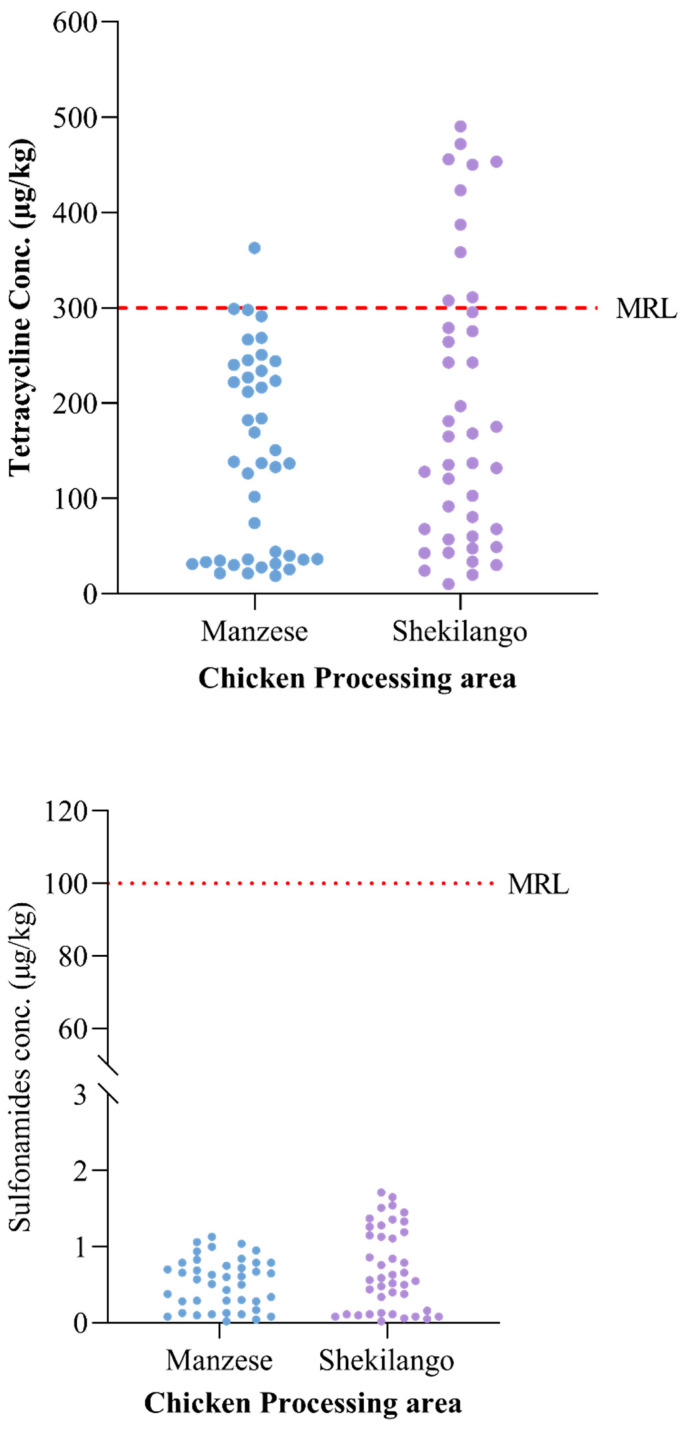
Maximum residue limit cut-off points for tetracycline and sulphonamide residues in chicken liver tissue samples.

**Figure 3 antibiotics-11-01222-f003:**
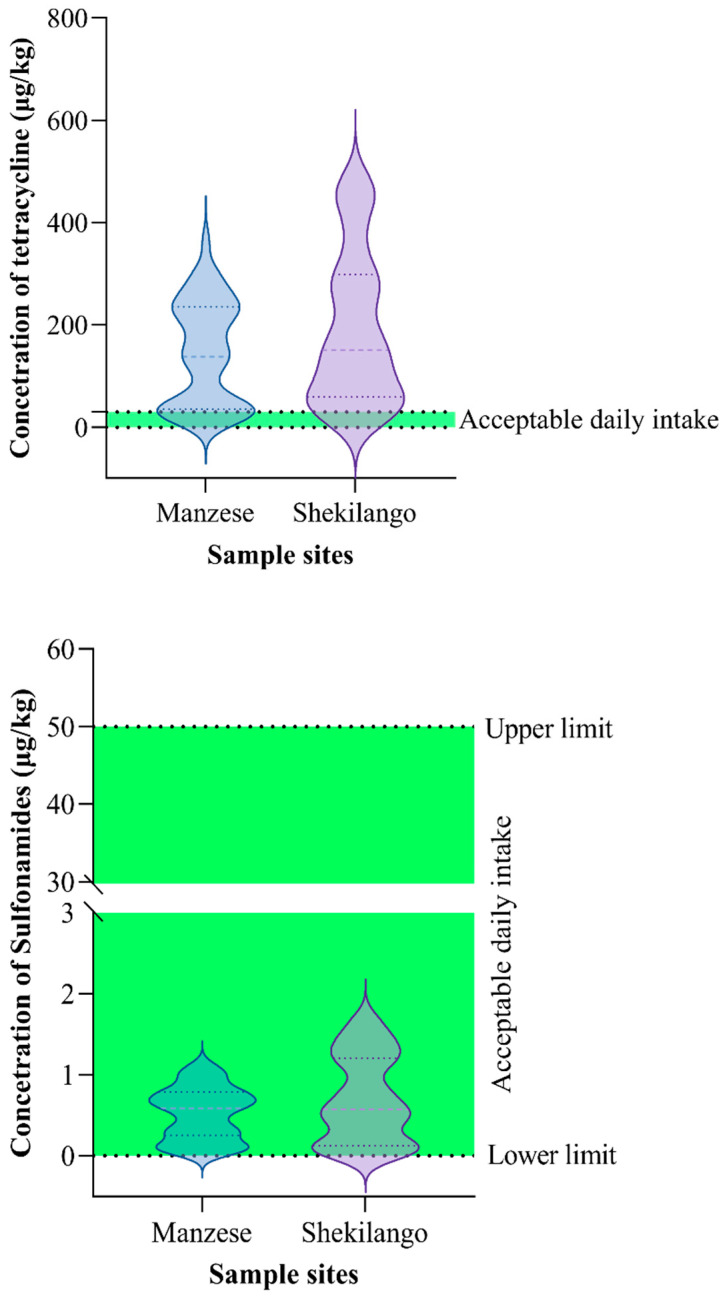
Comparison of concentrations of tetracycline and sulphonamide residues with acceptable daily intake (ADI).

**Table 2 antibiotics-11-01222-t002:** Median concentrations and interquartile ranges of drug residues.

	Median Concentration (IQR)		
Chicken Processing Area	Manzese	Shekilango	*p*-Value
Tetracycline	137.8 (35.48, 235.51)	151.06 (59.53, 298.50)	0.185
Sulphonamides	0.58 (0.25, 0.79)	0.58 (0.13, 1.21)	0.262

**Table 3 antibiotics-11-01222-t003:** The different compounds from sulphonamides and tetracycline and their corresponding cross-reactivity.

SULPHONAMIDES COMPOUNDS	CROSS-REACTIVITY (%)
Sulfamerazine	100
Sulfadiazine	130.2
Sulfamonomethoxine	181.8
Sulfamethoxydizine	194.8
Sulfamethazine	89.3
Sulfisomidine	70.6
Sulfadimethoxine	140.8
Sulfadoxine	132.1
Sulfamethoxazole	86.6
Sulfadimoxine	144.7
Phthalylsulfathiazole	73.9
Sulfamethizole	76.5
Sulfathiazole	103.3
Sulfaquinoxaline	59.4
Sulfamethoxypyridazine	108.6
Sulfaclozine	72.1
Sulfachloropyridazine	59.6
Sulfabenzoy	59.6
**TETRACYCLINE COMPOUNDS**	
Doxycycline	100
Tetracycline	150
Minocycline	92
Pyrithione	76
Chlortetracycline	75
Demethylchromycin	70
Oxytetracycline	83

## Data Availability

The data presented in this study are available upon reasonable request from the corresponding author.

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
