# Peer review of "Determination of Sulphonamides and Tetracycline Residues in Liver Tissues of Broiler Chicken Sold in Kinondoni and Ilala Municipalities, Dar es Salaam, Tanzania"

_antibiotics, 2022, doi:10.3390/antibiotics11091222_

Round 1
Reviewer 1 Report
The paper entitled “Determination of Sulphonamides and Tetracycline Residues in Liver Tissues of Broiler Chicken Sold in Kinondoni and Ilala Municipalities, Dar es Salaam, Tanzania” reported
cross-sectional study determined the concentration of sulphonamides and tetracycline residues in liver tissues of commercial broiler chicken sold in Dar es Salaam, Tanzania. Results showed that all 100% (n=84) samples had tetracycline residues and 21.4% (n=18) samples had sulphonamide residues, and 21.4% (n=18) had both sulphonamide and tetracycline residues. The observed presence of antibiotic residues in poultry tissues poses a health risk to consumers, and may lead to antimicrobial resistance micro-organisms, which may spread to humans and animals via the environment. This work is interesting and can be a good reference for readers of related topics. However, the method used for the risk assessment seems not a good choice for confirmation. Before the recommendation of publication, the following concerns should be addressed.
1. In the introduction, please provide reasons for the necessity of testing sulphonamides and tetracyclines in liver tissues of broiler chicken, rather than other food matrices.
2. Using the ELISA kit is usually a screening method without not quite an accurate quantification. Besides, the interference of similar compounds can not be avoided. Please provide the information on solving this problem.
3. Please provide the details in method validation of the ELISA kit, in terms of LOD, linear range, recovery, and stability.
4. There is no information on how you conclude that tetracycline residue exceeds acceptable daily intake (ADI).
5. There are many compounds belonging to sulphonamides and also at least three compounds possessing similar structure and function to tetracycline( chlortetracycline, oxytetracycline, doxycycline). Please provide which compounds have been tested positive with this ELISA method. How about their cross-reactivity?
6. Since a 6-month cross-sectional study had been conducted, what is the residue level profile of these compounds in different months?
7. In line 45, “the upper layer of N-hexane phase was removed, and 50 uL of deposit was taken for analysis”, how do you get the 50 uL of deposit, since only n-hexane had been added to dissolve the dried residue?
8. In line 60, “Tetracycline (TCs) (sensitivity 0.1 ppb, detection limit 4 ppm”, can you check the detection limit again, it seems too low a sensitivity.
Author Response
|
S/N |
REVIEWER 1: Comments and Suggestions for Authors |
Authors response |
||||||||||||||||||||||||||||||||||||||||||||||||||||||
|
1 |
In the introduction, please provide reasons for the necessity of testing sulphonamides and tetracyclines in liver tissues of broiler chicken, rather than other food matrices. |
The reason for the necessity of testing sulphonamides and tetracycline in liver tissues of broiler chicken, rather than other food matrices has been included in the introduction, “Since the liver tissue is where you would expect to find a high amount of residues due to its toxicifying function, we chose to examine the concentration of sulphonamides and tetracycline there rather than in other food matrices. Additionally, the broiler chicken liver and other internal parts are heavily consumed by the low-income population in Dar es Salaam.”
A previous study comparing antibiotic residues in chicken showed that samples from the liver and breast of broiler chicken contain various levels of residues of different antibiotics, being higher in the liver than in meat (El Tahir et al 2021). Furthermore, antibiotics can penetrate and remain in an animal liver tissues, especially if the initial dosage of the antibiotic is high or if the withdrawal time is not sufficient before slaughtering. For this reason we decided to choose the liver tissues. Besides, broiler chicken liver and other internal parts are heavily consumed by the low-income population in Dar es Salaam |
||||||||||||||||||||||||||||||||||||||||||||||||||||||
|
2 |
Using the ELISA kit is usually a screening method without not quite an accurate quantification. Besides, the interference of similar compounds cannot be avoided. Please provide the information on solving this problem. |
1. The ELISA kit has advanced to quantify the sample by comparing the optical density values to the standard curve to determine the concentration, despite being a screening method. The same approach was done by Yasmin EL Tahir et al,2021. 2. The ELISA Kits were designed and approved to be able to accommodate the issues of any interference from related compounds. 3. Because of its high specificity and sensitivity, the ELISA method has currently demonstrated to provide reliable results due to an antigen-antibody reaction. 4. The method is very effective because multiple analyses can be run simultaneously without tedious sample pre-treatment. Several screening and confirmatory analytical methods for the detection of antibiotic residues have been developed [14]. The immunoassay assay method has been widely used in the evaluation of antimicrobial residues in the food chain [15]. For example, Ramatla et al. [15] evaluated different methods for monitoring antimicrobial residues in meat samples. They concluded that the enzyme-linked immunosorbent assay (ELISA) method provided excellent sensitivity and selectivity for the determination of antibiotic residues in meat samples, and therefore, in this study, we opted for the ELISA method. ElTahir,Y.;Elshafie,E.I.; Asi, M.N.; Al-Kharousi, K.; Al Toobi, A.G.; Al-Wahaibi, Y.; Al-Marzooqi, W. Detection of Residual Antibiotics and Their Differential Distribution in Broiler Chicken Tissues Using Enzyme-Linked Immunosorbent Assay. Antibiotics 2021, 10, 1305. https://doi.org/10.3390/ antibiotics10111305 14. Ebrahimpour, B.; Yamini, Y.; Rezazadeh, M. A sensitive emulsification liquid phase microextraction coupled with on-line phase separation followed by HPLC for trace determination of sulfonamides in water samples. Environ. Monit. Assess. 2015, 187, 4162. [CrossRef] [PubMed] 15. Ramatla, T.; Ngoma, L.; Adetunji, M.; Mwanza, M. Evaluation of Antibiotic Residues in Raw Meat Using Different Analytical Methods. Antibiotics 2017, 6, 34. [CrossRef] [PubMed]
|
||||||||||||||||||||||||||||||||||||||||||||||||||||||
|
3 |
Please provide the details in method validation of the ELISA kit, in terms of LOD, linear range, recovery, and stability. |
These have now been provided. In brief; - The Limit of Detection (LOD) for Tetracycline ELISA kit was 4ppb, the recovery rate was 70% - 120%, linear range of 0ppb to 8.1ppb, and stability at 25oC. - The Limit of Detection (LOD) for sulphonamides ELISA kit was 1ppb, the recovery rate was 85± 25%, liner range of 1ppb to 20ppb, and stability at 25oC.
|
||||||||||||||||||||||||||||||||||||||||||||||||||||||
|
4 |
There is no information on how you conclude that tetracycline residue exceeds acceptable daily intake (ADI). |
The conclusion section has been revised to include the tetracycline residues on ADI |
||||||||||||||||||||||||||||||||||||||||||||||||||||||
|
5 |
There are many compounds belonging to sulphonamides and also at least three compounds possessing similar structure and function to tetracycline ( chlortetracycline, oxytetracycline, doxycycline). Please provide which compounds have been tested positive with this ELISA method. How about their cross-reactivity?
|
The compounds belonging to sulphonamides and tetracycline tested with ELISA kit and their cross-reactivity includes;
|
||||||||||||||||||||||||||||||||||||||||||||||||||||||
|
6 |
Since a 6-month cross-sectional study had been conducted, what is the residue level profile of these compounds in different months?
|
The design of this study was cross-sectional, implying that samples were not analyzed monthly. The actual sample collection and processing took only one month, November 2021. The 6-month study period includes the entire process, beginning with the development of the concept note. We have therefore rectified this in the manuscript |
||||||||||||||||||||||||||||||||||||||||||||||||||||||
|
7 |
In line 45, “the upper layer of N-hexane phase was removed, and 50 uL of deposit was taken for analysis”, how do you get the 50 uL of deposit, since only n-hexane had been added to dissolve the dried residue? |
Before adding 1ml of N-hexane, the dry residue was dissolved in 1ml of dilute redissolving solution. The information that is lacking has been added. |
||||||||||||||||||||||||||||||||||||||||||||||||||||||
|
8 |
In line 60, “Tetracycline (TCs) (sensitivity 0.1 ppb, detection limit 4 ppm”, can you check the detection limit again, it seems too low a sensitivity. |
The changes on the detection limit has been accommodated. Instead of 4 ppm, the detection limit was 4 ppb. |

Reviewer 2 Report
The presented study is only of very limited value as it has several severe flaws.
The first is the poor analysis method that does not allow exact determination of residues in chicken liver for sulphonamides and tetracycline.
ELISA method might be justifiable as a screening method but the exact residues have to be determined by HPLC/MS, this is the international standard.
Also, according to EMEA/MRL/023/95 the MRL for tetracycline in liver is 300 µg/kg and for residue analysis the 4-epimers of tetracyclines have to be determined as well (this lacking here).
Discussion:
This finding is in line with another study on poultry production in Dar es Salaam, which reported tetracycline to be the most used antibiotic by 49% followed by sulphonamides by 18% [7] and does indicate non-observation of withdrawal period [4].
This is not correct, there is no zero-tolerance with residues from liver concerning tetracyclines and sulfonsamides but rather, values have to be below MRL to slaughter the chicken, and it looks like (with this relatively unreliable analysis method) that none of the samples exceeded the MRL!
Thus this study does not give real indications of a quantifiable residue problem.
Author Response
|
S/N |
Reviewer 2: Comments and Suggestions for Authors
|
Authors response |
|
1 |
The first is the poor analysis method that does not allow exact determination of residues in chicken liver for sulphonamides and tetracycline. ELISA method might be justifiable as a screening method but the exact residues have to be determined by HPLC/MS, this is the international standard.
|
I agree that ELISA was once used as a screening method, but more recently, because of its high specificity and sensitivity (85 to 90%), ELISA has been used to quantify and qualify drug residue (Y Shahbazi et al 2015). In Tanzania, numerous studies have employed HPLC/MS, which are more time-consuming and expensive than ELISA. However, studies have shown that using ELISA can yield results that are comparable to those of HPLC/MS. The remaining antibiotics were found in broiler chicken tissues using the ELISA method in a recent study by Yasmin El Tahir et al. 2021. |
|
2 |
Also, according to EMEA/MRL/023/95 the MRL for tetracycline in liver is 300 µg/kg and for residue analysis the 4-epimers of tetracyclines have to be determined as well (this lacking here).
|
The JECFA-recommended MRL of 300 µg/kg for tetracycline and its classes in liver tissue. The 4-epimers of tetracycline were not included in the analysis, but they will be in subsequent research. We acknowledge this as a limitation and included in the appropriate section in the discussion. |
|
3 |
This finding is in line with another study on poultry production in Dar es Salaam, which reported tetracycline to be the most used antibiotic by 49% followed by sulphonamides by 18% [7] and does indicate non-observation of withdrawal period [4].
This is not correct, there is no zero-tolerance with residues from liver concerning tetracyclines and sulfonsamides but rather, values have to be below MRL to slaughter the chicken, and it looks like (with this relatively unreliable analysis method) that none of the samples exceeded the MRL!
|
JECFA's maximum residue limit of 300g/kg for tetracycline and its classes has been adopted. And this has resulted in having 13.1% of samples having tetracycline residues above the allowable maximum residue limit. |
|
4 |
Thus, this study does not give real indications of a quantifiable residue problem.
|
The approach used in the study of comparing the Optical density values obtained with the standard curve to determine the concentration of drug residues it give the indication of a quantifiable residues problem. ELISA results have been shown in studies to be comparable to those of HPLC/MS. In a recent study, Yasmin El Tahir et al. 2021 discovered the remaining antibiotics in broiler chicken tissues using the ELISA method. |

Round 2
Reviewer 1 Report
I believe this manuscript can be published after the second revision.